# A Deep Learning Framework with Explainability for the Prediction of Lateral Locoregional Recurrences in Rectal Cancer Patients with Suspicious Lateral Lymph Nodes

**DOI:** 10.3390/diagnostics13193099

**Published:** 2023-09-29

**Authors:** Tania C. Sluckin, Marije Hekhuis, Sabrine Q. Kol, Joost Nederend, Karin Horsthuis, Regina G. H. Beets-Tan, Geerard L. Beets, Jacobus W. A. Burger, Jurriaan B. Tuynman, Harm J. T. Rutten, Miranda Kusters, Sean Benson

**Affiliations:** 1Department of Surgery, Amsterdam UMC Location Vrije Universiteit Amsterdam, 1081 HV Amsterdam, The Netherlands; t.sluckin@amsterdamumc.nl (T.C.S.);; 2Cancer Center Amsterdam, Treatment and Quality of Life, 1081 HV Amsterdam, The Netherlands; 3Cancer Center Amsterdam, Imaging and Biomarkers, 1081 HV Amsterdam, The Netherlands; k.horsthuis@amsterdamumc.nl; 4Department of Radiology, Amsterdam UMC Location Vrije Universiteit Amsterdam, 1081 HV Amsterdam, The Netherlands; s.kol@amsterdamumc.nl; 5Department of Radiology, Catharina Hospital, 5623 EJ Eindhoven, The Netherlands; 6GROW School for Oncology & Developmental Biology, Maastricht University, 6211 LK Maastricht, The Netherlands; 7Department of Radiology, Netherlands Cancer Institute, 1066 CX Amsterdam, The Netherlands; 8Department of Clinical Radiology, University of Southern Denmark, Odense University Hospital, 5000 Odense, Denmark; 9Department of Surgery, Netherlands Cancer Institute, 1066 CX Amsterdam, The Netherlands; 10Department of Surgery, Catharina Hospital, 5623 EJ Eindhoven, The Netherlands; 11Department of Cardiology, Amsterdam University Medical Centers, University of Amsterdam, 1075 AX Amsterdam, The Netherlands

**Keywords:** lateral lymph nodes, rectal cancer, artificial intelligence, MR imaging, deep learning, explainability

## Abstract

Malignant lateral lymph nodes (LLNs) in low, locally advanced rectal cancer can cause (ipsi-lateral) local recurrences ((L)LR). Accurate identification is, therefore, essential. This study explored LLN features to create an artificial intelligence prediction model, estimating the risk of (L)LR. This retrospective multicentre cohort study examined 196 patients diagnosed with rectal cancer between 2008 and 2020 from three tertiary centres in the Netherlands. Primary and restaging T2W magnetic resonance imaging and clinical features were used. Visible LLNs were segmented and used for a multi-channel convolutional neural network. A deep learning model was developed and trained for the prediction of (L)LR according to malignant LLNs. Combined imaging and clinical features resulted in AUCs of 0.78 and 0.80 for LR and LLR, respectively. The sensitivity and specificity were 85.7% and 67.6%, respectively. Class activation map explainability methods were applied and consistently identified the same high-risk regions with structural similarity indices ranging from 0.772–0.930. This model resulted in good predictive value for (L)LR rates and can form the basis of future auto-segmentation programs to assist in the identification of high-risk patients and the development of risk stratification models.

## 1. Introduction

Traditionally, patients with low locally advanced rectal cancer (LARC) have been treated with the combination of neoadjuvant (chemo)radiotherapy (n(C)RT) and a total mesorectal excision (TME) procedure [1]. The incorporation of these procedures over the last few decades has caused overall local recurrence (LR) rates to drop to 5–10% [2]. However, patients with low LARC have an increased risk of spreading to lateral lymph nodes (LLNs), which surround the internal iliac and obturator vessels. Recent research shows that, if these LLNs are enlarged and treated inadequately, the lateral local recurrence (LLR) risk is significantly increased [3,4,5]. The treatment of LLR is challenging and associated with significant morbidity and mortality rates [2,3], with 5-year overall survival rates around 35% [4]. Unfortunately, the current proportion of LLRs is increasing and accounts for approximately 50% of all LRs [3,6]. This increase, compared to the relative reduction in overall LRs, suggests that current treatment of LLNs is insufficient. These high rates of recurrence, combined with the challenging treatment of recurrent disease, demand better primary identification of malignant LLNs.

One major problem is that specific diagnostic criteria for suspicious LLNs are limited, and it is largely unknown which features of LLNs are predictive of LLR. The Lateral Node Consortium study found that LLNs with a primary short-axis (SA) diameter of ≥7 mm resulted in a 5-year LLR rate of 19.5% [7]. Furthermore, Ogura et al., found that the incorporation of the size and location on the restaging MRI after neoadjuvant therapy was essential. Patients with an internal iliac LLN primarily ≥7 mm and that remained >4 mm on the restaging MRI resulted in a 52.3% LLR rate at 5 years [8]. These enlarged LLNs benefited from a lateral lymph node dissection (LLND). This procedure entails the complete removal of all lymphatic tissue from the lateral compartments and resulted in a 5-year LLR rate of only 6% [8]. These data not only indicate that the short-axis diameter of LLNs is most likely an important prognostic factor, but implies that an LLND may be crucial in the successful reduction of LLR rates in this patient group [9,10].

Considering these results, adequate awareness, knowledge, and treatment of malignant LLNs are essential. Presently, the main diagnostic criteria of size and location are assessed by radiologists. However, a recent study of 53 Dutch radiologists demonstrated significant inter-physician variation in the classification of the anatomical location and short-axis size of LLNs [11]. Similar research regarding mesorectal lymph nodes also found challenges in intra- and inter-observer agreement for mesorectal lymph node size, with low reproducibility of morphological characteristics [12]. These studies demonstrate the human factor involved in radiology, allowing for variation to occur during the evaluation of LLNs. To our knowledge, diagnostic criteria for suspicious LLNs in LARC patients using MRI scans have never been investigated before using deep learning AI methods, though similar deep learning models have explored nodal detection in other cancer types with promising results for clinical application [13,14,15,16].

This study aimed to create an AI prediction model with data from three expert centres to identify malignant LLNs with a higher risk of LLR. This incorporated the identification of imaging and clinical features that are associated with a high risk of (L)LR.

## 2. Materials and Methods

### 2.1. Study Population and Design

This multicentre retrospective cohort study included patients from three expert hospitals in the Netherlands: Amsterdam UMC, location VUmc (AUMC), the Netherlands Cancer Institute (NKI), and Catharina Hospital (CH). Patients from the NKI and CH were selected from the retrospective Lateral Node Consortium study [7], and patients from AUMC were included from a retrospective study of LARC patients [17]. Both studies only included patients with primary rectal carcinoma, of at least cT2 stage, diagnosed between July 2008 and November 2020. Patients with synchronous distant metastases or a non-curative dissection were excluded. None of the patients underwent formal LLND. Exclusion criteria were the absence of good-quality MRI scans unsuitable for segmentation, no restaging images, or missing clinical features required for the deep learning model (Figure 1).

The primary outcome was the prediction of (L)LR via AI models and was correlated with LLN features. For patients without surgery, the date of the primary MRI was used as the start of follow-up. All data were obtained from electronic health records. The study received central approval by the Institutional Review Board (IRB) of the Netherlands Cancer Institute on 2nd June 2021. Each participating centre reviewed the study protocol and provided approval. Informed consent was waived by the central IRB and by each participating centre during local ethical review and approval.

### 2.2. Initial MRI Re-Review

Primary and restaging MRIs of all patients were evaluated during the original studies by at least one expert radiologist in each centre. The expert radiologist in each centre was a specialized abdominal radiologist with vast experience and proven knowledge within colorectal assessment. Assessment of the LLNs was based on the largest node on the primary MRI and included the short- (SA) and long-axis (LA) diameter (measured on the transversal, coronal, or sagittal plane), the presence of malignant features (heterogeneity, border irregularity, loss of fatty centre, and shape), and anatomical location. During re-review, radiologists used a colour atlas to help determine the anatomical location of the LLN [8]. Lymphatic tissue located between the mesorectal fascia (MRF) and the lateral border of the main trunk of the internal iliac artery was considered the internal iliac compartment, and tissue ventral of the external iliac vessels was the external iliac compartment. The obturator compartment contained all lymphatic tissue lateral of the main trunk of the internal iliac artery, and once the internal iliac artery exited the pelvis, all remaining tissue was also regarded as the obturator compartment (Figure 2).

If an (L)LR occurred during follow-up, the imaging was re-assessed to define the exact location of the (lateral) local recurrence.

### 2.3. Image Segmentation

The LLNs documented during re-review were identified by the central researchers in order to establish a region of interest (RoI). In an attempt to avoid intra-observer bias and limit variation, two independent assessors evaluated the scans (T.C.S., M.H.). If the LLN was not easily identified, a senior researcher (M.K.) was contacted for advice. The RoI was created using segmentations (Figure 3) made on the transverse planes of T2-weighted MR imaging, with a maximum slice thickness of 5 mm. All visible LLNs were manually delineated on all slices of the MRI, checked by two researchers, and labelled on the primary and restaging MRI (Figure 3), using 3D slicer [18]. This segmentation distinguished the RoI from background tissue at the voxel level. This resulted in the creation of a binary mask with the same dimensions as the original image for each node.

### 2.4. Feature Extraction and Classification

A deep learning model was trained using a multi-channel convolutional neural network (CNN), with the MR images as the input for the CNN. Three-dimensional images were created for the primary and restaging MRI scans by resampling the transversal, coronal, and sagittal planes to an isotropic voxel size of 1 mm^3^. A 6 mm buffer in each direction of healthy tissue was created around the RoIs on the primary MRI. A multi-channel input image was then created consisting of the primary MRI, the restaging MRI, and the LLN RoIs. To create a standardised input size for the CNN, each RoI was resized using cubic spline interpolation to a 64 × 64 × 64 voxel volume.

The imaging model was trained using the in-house DTOR package, which uses the PyTorch framework [19]. The deep learning model was trained and adapted for the present LLN dataset by the implementation of transfer learning. Several backbone architectures, including both 2D and 3D models, were investigated to determine which performed best for the current dataset. For the case of 2D models, the axial slice with the largest LLN area from the manual segmentation was used. The 2D VGG19 backbone [20] was found to perform best, and a pre-trained version of the VGG19 model was used based on weights from the ImageNet dataset [21]. The intensities of the preprocessed RoIs were normalised to match the intensity distribution of the ImageNet data used for pretraining. During the subsequent transfer learning stage, model performance was optimised for the present dataset by the implementation of several optimisations. These consisted of early stopping (discontinuation of analyses when test performance did not improve), a decaying learning rate, a focal loss and class weighting (to account for imbalanced datasets), sharpness-aware minimisation (in which the smoothness of the loss function was taken into account to ensure convergence was not to an unphysical minimum) [22], and input data augmentations. The data augmentations consisted of random affine transformations, Gaussian blurring, and random rotations with a ten-degree standard deviation. Hyperparameter tuning was performed using 3-fold cross-validation, in which all of the patients were randomly assigned to be part of the test set of one of the folds. The first fold was used for hyperparameter tuning, during which a tree-structured Parzen estimator algorithm was applied. The tuned parameters included the learning rate decay, the focal loss parameters, the number of fixed layers of the pre-trained model, the early stopping threshold, and the learning rate. The optimised hyperparameters were used for the remaining 2 folds. Performance was found to be consistent between all three folds.

The model training with the tuned hyperparameters used patients from AUMC and CH, with the NKI forming a separate and independent test dataset. The trained model was used to calculate prediction scores of the risk on an (L)LR for each LLN. 

Model performance was assessed by calculating the area under the curve (AUC) of the receiver operating characteristics (ROC) curves for these prediction scores. The performance of the AI prediction model was also evaluated using the precision, sensitivity, and F1 scores, where the F1 score is defined as:F1 = 2 × (precision × sensitivity) / (precision + sensitivity)

For the case of the F1, sensitivity, and specificity results, the optimal cut-off point was chosen separately for the imaging and clinical models according to:TP × (1 − FP)
where TP and FP refer to the number of true positives and false positives, respectively.

For the trained models, a variety of explainability methods were used in order to provide local explanations of predictions. These methods included saliency compared to multiple different class-activation-mapping (CAM) methods [23]. This uses the gradients of any target concept, in our case the prediction of LR, flowing into the final convolutional layer to produce a coarse map, highlighting the important regions in the image for predicting (lateral) locoregional recurrence. The CAM methods included in this study were GradCAM, which weights 2D activations by the average gradient; FullGrad, which uses the sum of the gradients of biases from all over the network; XGradCAM, which uses gradients scaled according to the normalised activation functions; and EigenCAM, which uses the principal component of the activation functions. In order to compare the different explainability methods, the structural similarity index (SSIM) and mean-squared error (MSE) metrics were used. The SSIM metric is defined as
$$SSIM\left(x,y\right)=\frac{\left(2\mu_{x}^{2}\mu_{y}^{2}+c_{1})(2\sigma_{xy}+c_{2}\right)}{\left(\mu_{x}^{2}+\mu_{y}^{2}+c_{1})(\sigma_{x}^{2}+\sigma_{y}^{2}+c_{2}\right)},$$
where $\sigma_{x(y)}$ is the standard deviation of *x(y)*, $\mu_{x(y)}$ is the mean of *x(y)*, $\sigma_{xy}$ is the covariance between *x* and *y*, and the *c* constants are variables used to stabilise the division. Therefore, perfect agreement between two images would result in an SSIM of 1 and an MSE of 0.

### 2.5. Models

In total, three models were created, one based only on imaging features (Model 1), another based only on clinical features (Model 2), and a combined model with both imaging and clinical features from both sources (Model 3) (Figure 4). Imaging features included in Models 1 and 3 were the total LLN volume on baseline and restaging imaging (from manual segmentations) and the anatomical location of the LLN (according to MRI re-review). Models 2 and 3 included clinical information, such as the application of neoadjuvant treatment and, if so, which schedule was applied (chemoradiotherapy (25 × 2 Gy) with concomitant oral capecitabine or short-course radiotherapy (5 × 5 Gy)), the type of primary operation, and whether there were any pathologically positive resection margins (Table 1).

### 2.6. Statistical Analyses

Baseline analyses were performed using SPSS Statistics, Version 26.0 (SPSS, Chicago, IL, USA [24]). The AUC of ROC curves was calculated to assess the quality of the LLN features in the prediction of LR and LLR. For the case of the clinical model (no imaging features), the AdaBoost classifier [25] was used to determine the probability of (L)LR. The same features used in the clinical model were concatenated with the deep learning features to make a combined prediction model (Figure 5). 

## 3. Results

### 3.1. Patient Characteristics

A total of 427 patients diagnosed with LARC between July 2008 and November 2020 were eligible for the current study; however, 231 patients had to be excluded. This was due to no restaging MRI (*n* = 132), the absence of good-quality MRI scans sufficient for evaluation in the AI model (*n* = 60), or insufficient clinical data (*n* = 33) required for model development (Figure 1). This means that a total of 196 patients were included for analysis. The baseline characteristics are displayed in Table 1. The median follow-up time was 49 months (interquartile range (IQR) 21–70 months), and the total 4-year locoregional recurrence (LR) rate was 13.9% with an ipsi-lateral LR (LLR) rate of 5.5%. When only using the data derived from MRI re-review according to the short-axis size, AUCs of 0.57 (95% CI: 0.45–0.69) and 0.64 (95% CI: 0.47–0.81) were found for LR and LLR, respectively.

### 3.2. Deep Learning Model and Oncological Outcomes

An AUC of 0.67 (95% CI: 0.40–0.95) was found for the prediction of LR using the 2D VGG model described previously. For the case of LLR, this was determined to be 0.57 (95% CI: 0.046–1.00). Using only the clinical features described previously, the AUCs from the AdaBoost classifier were determined to be 0.68 (95% CI: 0.47–0.89) and 0.73 (95% CI: 0.46, 1.00) for LR and LLR, respectively. The use of the combined model, which was trained using both T2W MR imaging and clinical features, resulted in AUC values of 0.78 (95% CI: 0.60–0.96) and 0.80 (95% CI: 0.49–1.0) for LR and LLR, respectively. The ROC curve for the test dataset is shown in Figure 6. The full results of the AUC values per centre are shown in Table 2A,B for the LR and LLR endpoints, respectively.

### 3.3. Sensitivity and Specificity 

Using a database-generated, primary LLN short-axis size of ≥7 mm, as determined during MRI re-review, the sensitivity and specificity for developing an LR were 32.7% and 86.6%, respectively. When using imaging data from the current deep learning model, the corresponding sensitivities and specificities for the LR and LLR rates were 71.4% and 50.7%, which improved to 85.7% and 47.9% when also examining clinical features in combination with imaging features.

According to the database-generated ≥7 mm benchmark, the sensitivity and specificity for developing an LLR were 43.5% and 86.0%, respectively (Appendix A). When using imaging data from the current deep learning model, the corresponding sensitivities and specificities for the LR and LLR rates were 66.7% and 45.6%. When combining clinical and imaging features, this improved to 66.7% and 68.4%. The full results for each centre are provided in Table 3.

Similar improvements in sensitivity were seen for all versions of the deep learning model in the test centre, when only based on imaging or clinical features or a combination of both. 

### 3.4. Explainability

In order to test the explainability of the models, three examples from the test set were chosen in which the model was confident of a positive recurrence, unsure of the presence of a recurrence, and was confident no recurrence was present. These are depicted in Figure 7 along with the confidence values of the model. The results of the explainability methods described previously are shown in Figure 8. The corresponding SSIM and MSE values are provided in Table 4. Good agreement can be seen between the GradCAM, XGradCAM, and EigenCAM methods, with SSIM values ranging from 0.772–0.930 for these methods. The corresponding MSE values were found to be between 0.024 and 0.003. The saliency and FullGrad methods were found to show significantly less agreement. This is likely due to the pixel-based mapping for the case of saliency. For the case of FullGrad, the sum of gradients of all biases appeared to saturate the image.

## 4. Discussion

Given the increased risk of developing a locoregional recurrence due to either the insufficient treatment of or the failure to diagnose malignant lateral lymph nodes (LLNs), it is essential that malignant LLNs are adequately identified and treated appropriately. We created a deep learning model for the prediction of (L)LR rates in patients without an LLND, based on imaging features from segmented RoIs on T2W MRI images and combined these with clinical input features. This combined model was able to predict LR with an AUC of 0.78 and LLR with an AUC of 0.80, with a sensitivity of up to 85.7% for predicting LR and LLR in the test centre. However, while the sensitivity of local recurrence in the test set was high (85.7%), the corresponding specificity was found to be 47.9%. This is due to the class imbalance of the dataset and, as such, would be expected to improve with the use of larger datasets in future analyses.

The impact of AI in medical research has substantially increased over the past few years and is valuable for an accurate and standardised identification of patterns in medical data, as well as the combined assessment of biomedical images with integrated features, which are often too intricate for human discernment [26,27]. Artificial intelligence models have been used for the investigation of lymph node staging in colorectal cancer, where it was found that AI models outperformed radiologists in the assessment of lymph node metastasis [27]. Bedrikovetski et al., showed a pooled area under the ROC curve of 0.917 for deep learning models in the assessment of lymph node metastasis in rectal cancer and 0.808 for radiomics, both significantly higher than the AUC of 0.688 for the assessment by radiologists [27]. This is similar to the current results, where the AUCs for LR and LLR according to the deep learning models were 0.78 and 0.80, respectively, while the AUCs for LR and LLR according to short-axis measurements from radiologists were 0.57 and 0.64, respectively. 

The current study is, to our knowledge, one of only three studies to use deep learning models for the investigation of LLNs. The other two studies used either CT scans or excluded patients who underwent neoadjuvant therapy. This means that the current study, where segmented MRI scans were incorporated for the prediction of (L)LRs in patients treated with (chemo)radiotherapy without an LLND, is very appropriate for Western cohorts. This is in contrast to traditional Japanese protocols, where neoadjuvant treatment is usually not provided. Nakanishi et al. [28] and Kasai et al. [29] used an AI prediction model for pathologically enlarged LLNs related to pathological outcomes after an LLND. The radiomics model by Nakanishi et al., evaluated CT scans of 247 patients with rectal cancer and enlarged LLNs treated with n(C)RT and an LLND. The model was superior in the discrimination of malignant LLNs compared to the conventional diagnostic criteria by radiologists (AUC 0.91 vs.0.83, respectively) [30]. However, Nakanishi et al., used pre-treatment CT images for the segmentation and development of the prediction model, limiting the generalisability and external validity, considering that MRI is the standard staging method for rectal cancer [28]. Kasai et al., created a prediction model for a validation cohort of 56 patients with LLNs from MR images [29]. The AI prediction model was significantly more accurate in the prediction of recurrence compared to conventional methods using only the LLN SA size (AUC 0.85 versus 0.75, respectively). However, patients were excluded if they underwent neoadjuvant therapy, again limiting generalisability to a Western cohort. The current study has high external validity for Western patients by using MR images and patients who underwent neoadjuvant therapy. The AUC predictions of 0.78 and 0.80 for LR and LLR, respectively, are similar to those found by the aforementioned studies. 

In the current method, explainability images were created to identify areas of the pelvis most at risk for recurrence. These maps are aimed to aid radiologists in the identification and tracking of “at-risk” areas, to ensure thorough evaluation. This may be useful in situations where LLNs change shape or size, for example after neoadjuvant therapy, or when multiple suspicious LLNs are present. In such situations, it can be challenging to consistently ensure that the suspicious LLNs are visible on imaging series over time. This approach, to register image series to the baseline scan, would benefit the tracking of high-risk areas and provide more-accurate comparisons of changes over time and allow for this region to be closely monitored during intensive and frequent follow-up. This may be especially useful when LLNs are not well visualised on follow-up imaging series or when the pelvic anatomy has slightly changed since radiation and/or surgical treatment. Though lacking in supporting evidence, this interesting ability of the model should be explored further in the future. Our study indicates that standard saliency is potentially unsuitable for the identification of at-risk areas as residual disease is likely identified through combinations of voxels, while saliency considers the impact of individual voxels. The saturation effect of the FullGrad algorithm through the summation of bias gradients also appears to be unsuitable for the current use case.

This model was able to combine a select group of multicentre patients from three expert Dutch centres with LLNs to create prediction scores for locoregional recurrence. This can in the future be used as a foundation for larger, more-detailed, machine learning models for this patient population. Considering the wide range of MRI fields used, this model appears to be rather robust. This study was also able to support the limited number of similar studies, showing that deep learning AI models can predict LR and LLRs relatively accurately based on imaging and clinical features, with a high sensitivity. Future research can apply this model to multiple and diverse validation cohorts to accumulate new information and prediction scores. This can then, in turn, aid the creation of risk stratifications, prediction models, and auto-segmentation networks. The strengths of this study lie in the future possibilities. Similar deep learning models can be created for the auto-segmentation of LLNs, which would not only decrease the chances of inter-variability between human observers, but allow for automated identification of LLNs, which could assist radiologists in their diagnostic process. Furthermore, auto-segmentation would increase the sensitivity of similar models and provide more-refined feature identification and selection of RoIs, improving the overall input channels [27,31]. This would also allow for a start-to-finish AI-based situation, decreasing the chances of human error or inter-physician variation. However, the logistical and theoretical challenges when implementing an AI model for clinical use should not be overlooked or underestimated. A broad foundation of support would be necessary within the medical community, especially between radiologists, with an understanding of what the model represents and how it can aid the diagnostic process. 

This study had some limitations. Firstly, the current model required both primary and restaging imaging features for the development of prediction scores, which meant that many patients without restaging images had to be excluded. There was also heterogeneity present in patients between centres; however, the current model did perform the best for the test centre, which is reassuring. Another limitation was that some MRI scans differed in quality, limiting precision and accuracy during segmentation, and it Is possible that, because this study included retrospective data from different hospitals, different MRI protocols were adhered to, which might affect the obtained results. Furthermore, the current sample is rather limited, which may be reflected in the found sensitivity data. With a larger dataset, the sensitivity results could be largely improved. Even though two independent assessors evaluated the scans, in an attempt to limit intra-observer bias and variation, manual segmentation may still have been affected and auto-segmentation of LLNs with feature classification would have been ideal. Lastly, besides T2-weighted MRI, no other series, such as diffusion-weighted images, were incorporated into the model to further influence accuracy in recurrence predictability. Lastly, due to the fact that patients did not undergo an LLND, the clinical outcome and model development could not be linked to the pathological results.

## 5. Conclusions

A deep learning model was trained to predict the local recurrence and ipsi-lateral local recurrence risk based on magnetic resonance imaging and clinical features of 196 patients with lateral lymph nodes in rectal cancer. The predictive model using T2W imaging and clinical features resulted in AUC scores of 0.78 and 0.80 for LR and LLR, respectively. Explainability methods were found to identify the high-risk regions of the input, with a high level of consistency between the different methods. Future research is needed to validate the model in external cohorts and to develop new research directions such as auto-segmentation and risk stratification. 

## Figures and Tables

**Figure 1 diagnostics-13-03099-f001:**
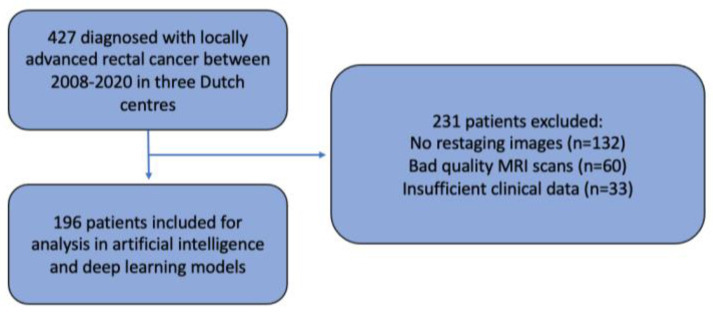
Patient selection.

**Figure 2 diagnostics-13-03099-f002:**
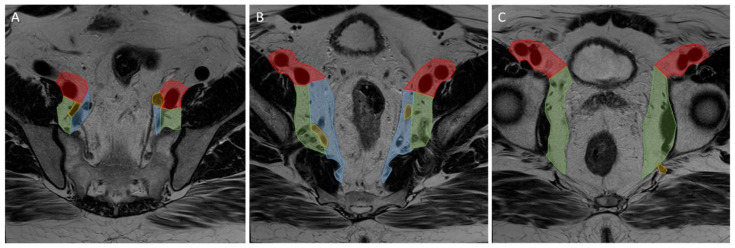
Lateral lymph node compartments. Progression through a T2 transversal pelvic MRI scan from (**A**–**C**) cranial to caudal. Red: external iliac compartment, blue: internal iliac compartment, green: obturator compartment, yellow: internal iliac artery.

**Figure 3 diagnostics-13-03099-f003:**
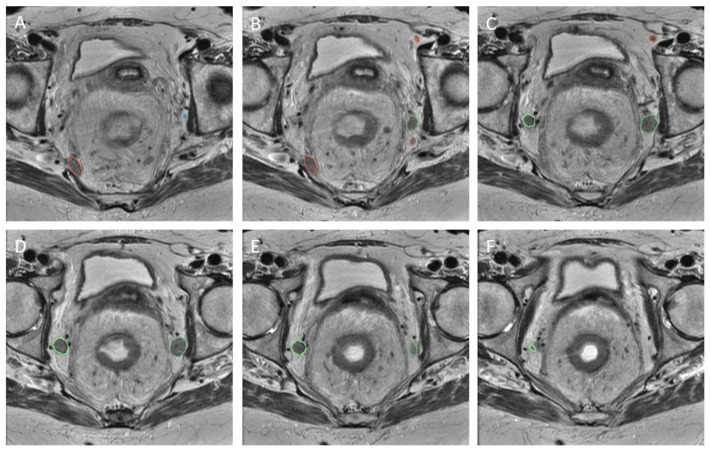
Segmentation. Caudal progression through T2 transversal MRI scan (from **A**–**F**). Five labelled LLNs.

**Figure 4 diagnostics-13-03099-f004:**
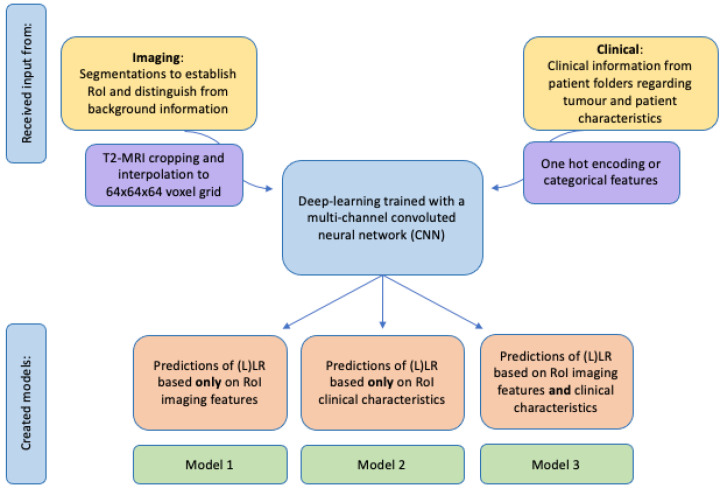
Schematic creation of prediction models with input from imaging and/or clinical sources.

**Figure 5 diagnostics-13-03099-f005:**
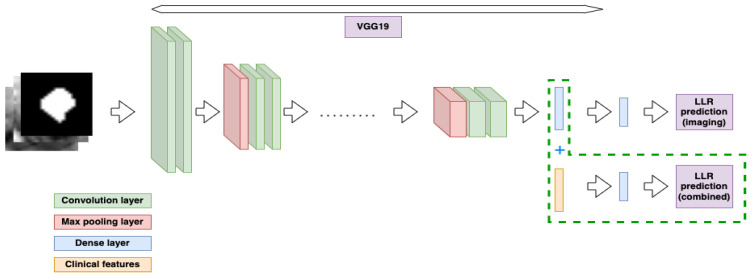
Diagram representing the deep learning model and the combination with clinical features.

**Figure 6 diagnostics-13-03099-f006:**
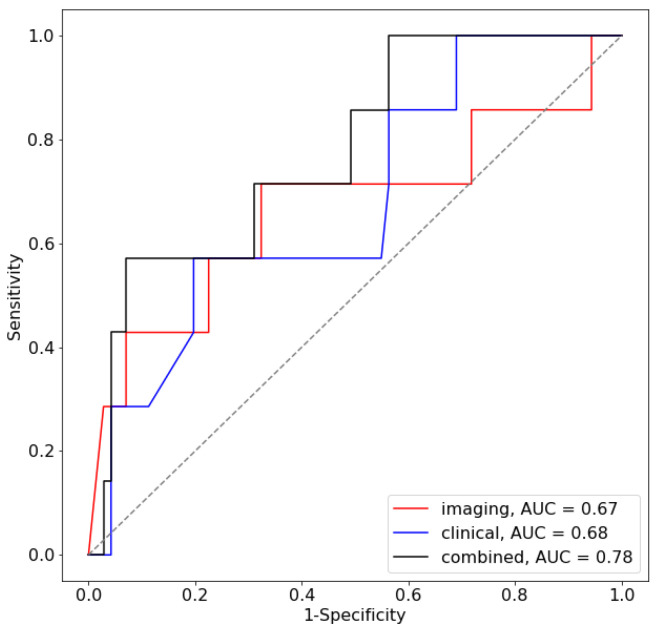
ROC curve for the prediction of LR for the test centre.

**Figure 7 diagnostics-13-03099-f007:**
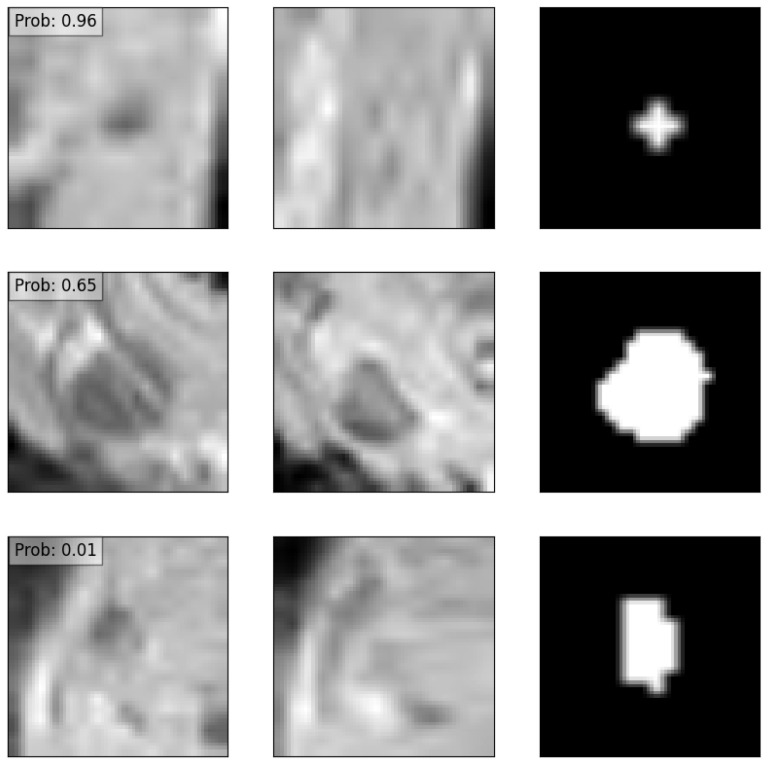
Three examples of an input data point corresponding to a positive recurrence with high confidence (**top row**), positive recurrence with low confidence (**middle row**), and no predicted recurrence with high confidence (**bottom row**). The left column depicts the node before treatment, the middle column after treatment, and the right column the manual delineation. Overlaid also are the predicted probabilities of recurrence provided by the deep learning model.

**Figure 8 diagnostics-13-03099-f008:**
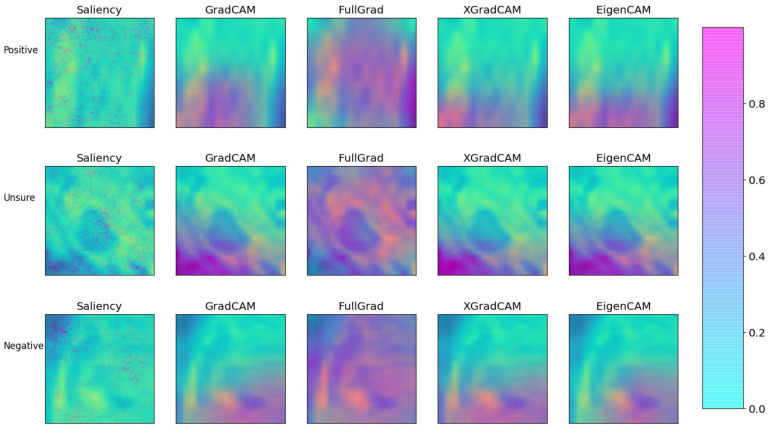
Comparison of the various explainability methods as defined in the text. Each method was applied to the three example cases of Figure 7. Each image was created by first plotting the post-treatment channel with the corresponding local attribution values subsequently added with fifty percent transparency.

**Table 1 diagnostics-13-03099-t001:** Baseline characteristics.

*N = 196*	*N (%)*
Male	127 (64.8)
Female	69 (35.2)
Age in years, mean (SD)	64.1 (10.8)
BMI, mean (SD)	26.0 (4.9)
Centre	
Catharina Hospital (CH)	116 (59.2)
Netherlands Cancer Institute (NKI)	24 (12.2)
Amsterdam UMC (AUMC)	56 (28.6)
ASA-score	
1	27 (15.3)
2	128 (72.7)
3	20 (11.4)
4	1 (0.6)
Distance tumour to anorectal junction in cm, mean (SD)	3.0 (2.8)
Clinical T-stage	
cT2	5 (2.6)
cT3	113 (57.7)
cT4	78 (39.8)
Clinical N-stage	
cN0	48 (24.5)
cN1	77 (39.3)
cN2	71 (36.2)
Positive mesorectal fascia or T4 on primary MRI	99 (50.5)
Anatomical location of largest lateral lymph node *	
Internal iliac	19 (9.7)
External iliac	18 (9.2)
Obturator	159 (81.1)
Mean lateral lymph node size on primary MRI, mm (SD) *	5.5 (2.7)
Mean number of lateral lymph nodes on primary MRI (SD)	3.6 (2.1)
Neoadjuvant treatment *	
Short-course radiotherapy	34 (17.3)
Chemoradiotherapy	162 (82.7)
Operation	
No surgery/wait and see	6 (3.1)
TEM/local excision	1 (0.5)
TME/LAR	108 (55.1)
APR	79 (40.3)
Pelvic exenteration	2 (1.0)
Lateral lymph node dissection (LLND)	
No	186 (94.9)
LLND	3 (1.5)
Node-picking	7 (3.6)
Positive resection margins *	15 (7.7)

Abbreviations: SD: standard deviation, AJN: anorectal junction, MRF: mesorectal fascia, RT: radiotherapy, TEM: transanal endoscopic microsurgery, TME: total mesorectal excision, LAR: low anterior resection, APR: abdominal perineal resection. * Features used for input in clinical and combined model.

**Table 2 diagnostics-13-03099-t002:** Performances of the different models for the prediction of LR (A) and LLR (B).

Centre	AUC (Imaging)	AUC (Clinical)	AUC (Combined)
A: local recurrence			
NKI	0.67 (95% CI: 0.40–0.95)	0.68 (95% CI: 0.47–0.89)	0.79 (95% CI: 0.60–0.96)
AUMC	0.85 (95% CI: 0.75–0.95)	0.82 (95% CI: 0.70–0.93)	0.68 (95% CI: 0.53–0.83)
CH	0.60 (95% CI: 0.53–0.67)	0.79 (95% CI: 0.73–0.84)	0.50 (95% CI: 0.43–0.58)
B: lateral local recurrence			
NKI	0.57 (95% CI: 0.46–1.00)	0.73 (95% CI: 0.46–1.00)	0.80 (95% CI: 0.49–1.00)
AUMC	0.82 (95% CI: 0.70–0.94)	0.81 (95% CI: 0.70–0.93)	0.61 (95% CI: 0.44–0.78)
CH	0.64 (95% CI: 0.56–0.71)	0.78 (95% CI: 0.71–0.84)	0.53 (95% CI: 0.45–0.62)

**Table 3 diagnostics-13-03099-t003:** F1, precision, and recall scores for the models, given in the order imaging (1st number), clinical (2nd number), and combined (3rd number) for the prediction of LR (A) and LLR (B).

Centre	F1	Specificity	Sensitivity
A: local recurrence			
NKI	0.21/0.24/0.24	50.7%/67.6%/47.9%	71.4%/57.1%/85.7%
AUMC	0.20/0.25/0.16	51.7%/79.4%/44.4%	91.7%/58.3%/83.3%
CH	0.26/0.43/0.22	59.1%/71.0%/57.2%	47.8%/71.6%/41.8%
B: lateral local recurrence			
NKI	0.11/0.17/0.13	45.6%/68.4%/52.6%	66.7%/66.7%/66.7%
AUMC	0.18/0.21/0.14	53.4%/77.9%/45.5%	88.9%/55.6%/77.8%
CH	0.28/0.42/0.24	58.6%/72.7%/58.6%	53.7%/66.7%/44.4%

**Table 4 diagnostics-13-03099-t004:** Structural similarity indices and mean-squared error values in the comparison of various explainability methods to standard GradCAM.

Case	Saliency	GradCAM	FullGRAD	XGradCAM	EigenCAM
SSIM					
Positive	0.235	1.000	0.225	0.772	0.792
Unsure	0.220	1.000	0.233	0.839	0.912
Negative	0.271	1.000	0.266	0.930	0.916
MSE					
Positive	0.214	0.000	0.281	0.024	0.024
Unsure	0.245	0.000	0.347	0.014	0.003
Negative	0.212	0.000	0.346	0.010	0.008

## Data Availability

Data can be provided upon reasonable request to the corresponding author.

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
