# Peer review of "A Deep Learning Framework with Explainability for the Prediction of Lateral Locoregional Recurrences in Rectal Cancer Patients with Suspicious Lateral Lymph Nodes"

_diagnostics, 2023, doi:10.3390/diagnostics13193099_

Round 1
Reviewer 1 Report
This manuscript classifies the cropped LLN of MRI images according to diagnostic categories of medical records using CNN.
1. What are the categories for LLN progressions available on the medical records?
2. Using this CNN classification framework, the obtained sensitivity and specificity are 85,7% and 67,6%, respectively. In my opinion, this framework should achieve a higher value for these metrics.
3. The explanation of Figure 3 seems missing in the paragraph.
-
Author Response
1. What are the categories for LLN progressions available on the medical records? -> There are no categories for progression; from the medical records initial and restaging sizes were recorded as well location. These sizes were used when entering clinical data and compared to the volumes calculated by the segmentations.
2. Using this CNN classification framework, the obtained sensitivity and specificity are 85,7% and 67,6%, respectively. In my opinion, this framework should achieve a higher value for these metrics. -> The performance is dependent on the framework, the initial dataset that is the basis of the transfer learning, and the dataset that is the target of the training. It is expected that with more patients and with the use of the DWI sequence that the performance would be better. We have mentioned this as a limitation of the study.
3. The explanation of Figure 3 seems missing in the paragraph. ->Extra details regarding this process have been added to lines 137/138.
Reviewer 2 Report
In this paper, authors explored LLN features to create an artificial-intelligence prediction-model, estimating the risk of (L)LR. This retrospective multicentre cohort study examined 196 patients diagnosed with rectal cancer between 2008-2020 from three tertiary centres. Primary and restaging magnetic resonance imaging were used. Visible LLNs were segmented and used for a multi-channel Convolutional Neural Network. A deep-learning model was developed and trained for the prediction of (L)LR according to malignant LLNs. The authors did good work and interested for the readers. The following review comments are recommended, and the authors are invited to explain and modify.
1 The title does not make sense; it needs to be revised.
2 The main contributions of the manuscript are not clear. The main contributions of the article must be very clear and would be better if summarize them into 3-4 points at the end of the introduction.
3 The introduction section needs to be improved. An introduction is an important road map for the rest of the paper that should be consist of an opening hook to catch the researcher's attention, relevant background study, and a concrete statement that presents main argument but your introduction lacks these fundamentals, especially relevant background studies. This related work is just listed out without comparing the relationship between this paper's model and them; only the method flow is introduced at the end; and the principle of the method is not explained. To make soundness of your study must include these latest related works.
I (2022). Automatic interpretation and clinical evaluation for fundus fluorescein angiography images of diabetic retinopathy patients by deep learning. British Journal of Ophthalmology, 2022-321472. doi: 10.1136/bjo-2022-321472
II (2022). New insights into natural products that target the gut microbiota: Effects on the prevention and treatment of colorectal cancer. Frontiers in pharmacology, 13, 964793. https://doi.org/10.3389/fphar.2022.964793
III (2022). Limonin Exerts an Anti-Inflammatory Effect through the ERK/MEK Signaling Pathway in Colorectal Cancer. Journal of Biological Regulators and Homeostatic Agents, 36(4), 973-984. doi: 10.23812/j.biol.regul.homeost.agents.20223604.108
IV (2023). Analysis and Design of Surgical Instrument Localization Algorithm. Computer Modeling in Engineering & Sciences, 137(1), 669-685. doi: 10.32604/cmes.2023.027417
4 “This segmentation distinguished the RoI from background tissue”, what kind of background tissues? Need details of segmentation process using 3D slicer?
5 A deep-learning model was trained using multi-channel CNN, but how to do image preprocessing step?
6 The model training with the tuned hyperparameters, how to optimize these hyperparameters during model training?
7 When writing phrases like “Performance of the AI-prediction model was also evaluated using the precision, sensitivity, and F1 scores”, it must cite related works in order to sustain the statement 10.1155/2022/2665283; 10.1155/2023/2345835.
8 TP and FP refer to the number of true positives and false positives; confusion matrix needs to be shown.
9 Authors should mention the implementation challenges.
10 Moreover, it should be noticed that the clinical appliance has to be decided by medicals since the existing differences between the real image and the one generated by the proposed model could be substantial in the medical field.
Minor editing of English language required.
Author Response
1 The title does not make sense; it needs to be revised. -> this has been amended and hopefully clearer in this manner
2 The main contributions of the manuscript are not clear. The main contributions of the article must be very clear and would be better if summarize them into 3-4 points at the end of the introduction. -> the introduction has been checked for grammar by two native English speakers (1st and last author) and additional text has been added to elaborate on the main contributions
3. The introduction section needs to be improved. An introduction is an important road map for the rest of the paper that should be consist of an opening hook to catch the researcher's attention, relevant background study, and a concrete statement that presents main argument but your introduction lacks these fundamentals, especially relevant background studies. This related work is just listed out without comparing the relationship between this paper's model and them; only the method flow is introduced at the end; and the principle of the method is not explained. To make soundness of your study must include these latest related works.
I (2022). Automatic interpretation and clinical evaluation for fundus fluorescein angiography images of diabetic retinopathy patients by deep learning. British Journal of Ophthalmology, 2022-321472. doi: 10.1136/bjo-2022-321472
II (2022). New insights into natural products that target the gut microbiota: Effects on the prevention and treatment of colorectal cancer. Frontiers in pharmacology, 13, 964793. https://doi.org/10.3389/fphar.2022.964793
III (2022). Limonin Exerts an Anti-Inflammatory Effect through the ERK/MEK Signaling Pathway in Colorectal Cancer. Journal of Biological Regulators and Homeostatic Agents, 36(4), 973-984. doi: 10.23812/j.biol.regul.homeost.agents.20223604.108
IV (2023). Analysis and Design of Surgical Instrument Localization Algorithm. Computer Modeling in Engineering & Sciences, 137(1), 669-685. doi: 10.32604/cmes.2023.027417
-> additional text has been added to make the introduction clearer. However, the 4 references mentioned do not relate to the current topic, only in some way to AI, which is why we do not believe that they improve the introduction.
4 “This segmentation distinguished the RoI from background tissue”, what kind of background tissues? Need details of segmentation process using 3D slicer?
-> This refers to any non-nodal tissue. More details have been added.
5 A deep-learning model was trained using multi-channel CNN, but how to do image preprocessing step?
-> The preprocessing consisted of the interpolation of the different sequences and the extraction of the RoIs. The normalization was taken from the ImageNet pretraining. This detail has been added.
6 The model training with the tuned hyperparameters, how to optimize these hyperparameters during model training?
-> The hyperparameters are optimized before the training, with the use of the Parzen estimator on the validation set of the first fold, with the best performing set being used for the training of the other 2 folds. This has been reworded to make clearer.
7 When writing phrases like “Performance of the AI-prediction model was also evaluated using the precision, sensitivity, and F1 scores”, it must cite related works in order to sustain the statement 10.1155/2022/2665283; 10.1155/2023/2345835.
-> Precision and sensitivity are standard in the literature, as is F1 score. However, the latter has been defined for clinical readers.
8 TP and FP refer to the number of true positives and false positives; confusion matrix needs to be shown.
-> We have included the confusion matrix.
9 Authors should mention the implementation challenges.
-> Sentences in lines 380-386 have been added to ensure that we understand the implementation challenges and that readers contemplate these as well.
10 Moreover, it should be noticed that the clinical appliance has to be decided by medicals since the existing differences between the real image and the one generated by the proposed model could be substantial in the medical field.
-> this has also been added to the sentences in 380-386 to reflect this important point
Reviewer 3 Report
Thank you for the opportunity to review this paper.
Overall, a very interesting and well-written paper.
Introduction
Please include a reference to this statement “Recent research has demonstrated that 50 the short-axis (SA) diameter is an important prognostic factor.”
introduction is lacking perspectives on deep learning models. The authors can consider including short perspectives on this.
A bit vague perspective “ Similar deep-learning models have explored nodal detection in other cancer types with 70 promising results (13-16).”
Methods
Were the patients scanned with the exact same MRI protocol and the same kind of MR tesla unit?
If not, consider providing information on the MRI protocols.
What is the definition of an expert radiologist?
Table 1. Why only includes males? Why not report from both males and females?
Nowadays it seems a bit old fashion not to include to both sexes equally.
Discussion
Perspectives on implementation in the future? Can AI be used?
Discussion on sensitivity can be included as it was often low.
Limitation
The section can benefit from including perspectives on how various MRI protocols may affect AI results.
Did various scan protocols and MRI manufacturers affect the study results?
Author Response
Introduction
Please include a reference to this statement “Recent research has demonstrated that 50 the short-axis (SA) diameter is an important prognostic factor.”
-> the sentence has been changed so that the reference (7) is clear
introduction is lacking perspectives on deep learning models. The authors can consider including short perspectives on this.
-> lines 76/77 reflect on the available evidence of deep learning models recently studied within oncological medical research.
A bit vague perspective “ Similar deep-learning models have explored nodal detection in other cancer types with 70 promising results (13-16).”
-> this has been changed somewhat to be clearer.
Methods
Were the patients scanned with the exact same MRI protocol and the same kind of MR tesla unit?
If not, consider providing information on the MRI protocols.
-> the protocols are not available anymore as the data scans from 2008-2020, but we have added this as a limitation in lines 392-394.
What is the definition of an expert radiologist?
-> this has been added 114/115
Table 1. Why only includes males? Why not report from both males and females?
Nowadays it seems a bit old fashion not to include to both sexes equally.
-> by providing the percentage of males it is easily deductible what percentage are females. However, we understand your point and to avoid any issues the percentage of females has been added to table 1.
Discussion
Perspectives on implementation in the future? Can AI be used?
-> please see additional sentences in lines 380-386
Discussion on sensitivity can be included as it was often low.
-> Specificity was often low due to the class imbalance of the dataset. This is a very good point. As we mention we did introduce a class weighting in the loss function, but this can only help so much. We have added some extra discussion.
Limitation
The section can benefit from including perspectives on how various MRI protocols may affect AI results.
Did various scan protocols and MRI manufacturers affect the study results?
-> A sentence has been added to line 392/393 to reflect this.
Round 2
Reviewer 2 Report
Accept